# Safety Evaluation of Employing Temporal Interference Transcranial Alternating Current Stimulation in Human Studies

**DOI:** 10.3390/brainsci12091194

**Published:** 2022-09-05

**Authors:** Yi Piao, Ru Ma, Yaohao Weng, Chuan Fan, Xinzhao Xia, Wei Zhang, Ginger Qinghong Zeng, Yan Wang, Zhuo Lu, Jiangtian Cui, Xiaoxiao Wang, Li Gao, Bensheng Qiu, Xiaochu Zhang

**Affiliations:** 1Application Technology Center of Physical Therapy to Brain Disorders, Institute of Advanced Technology, University of Science & Technology of China, Hefei 230031, China; 2Department of Radiology, The First Affiliated Hospital of USTC, Hefei National Research Center for Physical Sciences at the Microscale and School of Life Science, Division of Life Science and Medicine, University of Science & Technology of China, Hefei 230027, China; 3Centers for Biomedical Engineering, School of Information Science and Technology, University of Science and Technology of China, Hefei 230027, China; 4Department of Psychiatry, The First Affiliated Hospital of Anhui Medical University, Hefei 230022, China; 5School of Optometry and Vision Science, Cardiff University, Cardiff CF24 4HQ, UK; 6SILC Business School, Shanghai University, Shanghai 201800, China; 7Department of Psychology, School of Humanities & Social Science, University of Science and Technology of China, Hefei 230027, China; 8Biomedical Sciences and Health Laboratory of Anhui Province, University of Science & Technology of China, Hefei 230027, China

**Keywords:** TI-tACS, safety, noninvasive brain stimulation, transcranial electric stimulation, EEG, TI, tACS

## Abstract

Temporal interference transcranial alternating current stimulation (TI-tACS) is a new technique of noninvasive brain stimulation. Previous studies have shown the effectiveness of TI-tACS in stimulating brain areas in a selective manner. However, its safety in modulating human brain neurons is still untested. In this study, 38 healthy adults were recruited to undergo a series of neurological and neuropsychological measurements regarding safety concerns before and after active (2 mA, 20/70 Hz, 30 min) or sham (0 mA, 0 Hz, 30 min) TI-tACS. The neurological and neuropsychological measurements included electroencephalography (EEG), serum neuron-specific enolase (NSE), the Montreal Cognitive Assessment (MoCA), the Purdue Pegboard Test (PPT), an abbreviated version of the California Computerized Assessment Package (A-CalCAP), a revised version of the Visual Analog Mood Scale (VAMS-R), a self-assessment scale (SAS), and a questionnaire about adverse effects (AEs). We found no significant difference between the measurements of the active and sham TI-tACS groups. Meanwhile, no serious or intolerable adverse effects were reported or observed in the active stimulation group of 19 participants. These results support that TI-tACS is safe and tolerable in terms of neurological and neuropsychological functions and adverse effects for use in human brain stimulation studies under typical transcranial electric stimulation (TES) conditions (2 mA, 20/70 Hz, 30 min).

## 1. Introduction

Brain stimulation techniques are widely used in neurology and psychiatry research and treatments. In recent decades, noninvasive brain stimulations, such as transcranial direct current stimulation (tDCS), transcranial alternating current stimulation (tACS) and transcranial magnetic stimulation (TMS), have been highly regarded since they are not as costly as deep brain stimulation (DBS), which requires surgery and causes injury [1,2]. However, current noninvasive brain stimulations cannot modulate deep brain areas such as the hippocampus. Therefore, a noninvasive deep brain stimulation technique is desired. Fortunately, Grossman et al. proposed temporal interference transcranial alternating current stimulation (TI-tACS) in 2017, which can mediate the activation of hippocampal neurons in mice without eliciting activation in other cortical areas [3]. Conventional tACS takes effect directly with its low-frequency current, which is easy to drop off and will activate the superficial brain areas first. But TI-tACS takes effect indirectly with an envelope that is generated by two electric fields at high frequencies with a small frequency difference (e.g., 2 kHz and 2.02 kHz). The two high-frequency currents per se cannot entrain neural electrical activities due to the low-pass filtering feature of the neural membrane. However, at the intersection of the two currents, there would form an envelope at a low-frequency equal to the difference of the two high-frequency currents (e.g., 20 Hz), which can modulate deep brain areas like conventional tACS (Figure 1) [4,5]. Although the results of subsequent animal studies [6,7,8], simulation studies [9,10,11,12,13,14] and human studies [15,16] supported the effectiveness of TI-tACS in stimulating brain areas in a selective manner, its safety in stimulating human brains is still unclear. It is necessary to verify the safety of TI-tACS before we apply TI-tACS to human participants and patients. Transcranial electric stimulation (TES) studies conventionally define safe stimulation conditions as stimulation currents with intensities of 1–2 mA and frequencies of 0–10 kHz and stimulation durations of 10–30 min/day [17]. Although some studies have tried to test the safety of TES in extreme conditions, such as with a current of 4 mA or a duration of 60 min, as a pioneering study, we think it is necessary to test the safety of TI-tACS under conditions similar to typical stimulation conditions (2 mA, 20/70 Hz, 30 min). We hypothesized that TI-tACS is safe and generally well-tolerated under these stimulation conditions.

In the current study, we implemented TI-tACS or sham stimulation on the left primary motor cortex (M1) of healthy participants to validate the safety of TI-tACS in the human brain. In addition, key neurological and neuropsychological measures and electroencephalograph (EEG) data were recorded before and after active or sham TI-tACS. Specifically, serum neuron-specific enolase (NSE) was measured as the biological marker of neuronal damage [18]; the Montreal Cognitive Assessment (MoCA) [19], an abbreviated version of the California Computerized Assessment Package (A-CalCAP) [20] and a self-assessment scale (SAS) were employed to evaluate participants’ cognitive functions and states; a revised version of the Visual Analog Mood Scale (VAMS-R) was employed to evaluate participants’ mood states [21,22]; the Purdue Pegboard Test (PPT) was employed to test participants’ motor function [23]; and participants also completed a questionnaire about adverse effects (AEs) [24,25].

## 2. Materials and Methods

### 2.1. Participants

Forty healthy adult participants were recruited in this study and were randomly assigned to the active group and the sham group. Two participants were removed from the analyses because of instrument and program failures. The remaining 38 participants were included in the following data analyses (active group: N = 19, 7 females, mean age ± SD: 22.00 ± 2.082 years, mean education level ± SD: 16.32 ± 1.797 years, mean handedness score ± SD: 78.42 ± 20.887; sham group: N = 19, 8 females, mean age ± SD: 23.26 ± 2.864 years, mean education level ± SD: 16.79 ± 2.123 years, mean handedness score ± SD: 84.74 ± 19.824). All participants were right-handed as assessed with the Edinburgh Handedness Inventory [26] and had normal or corrected-to-normal vision. No participant reported a history of craniotomy or injury to the head, personal or family history of neurological or psychiatric disease, metal implants or implanted electronic devices, skin sensitivity or use of medicine during the experiment. For safety reasons, volunteers who were pregnant or could be pregnant were rejected. Informed consent was obtained prior to any involvement in the study. This study was approved by the Human Ethics Committee of University of Science and Technology of China (IRB Number: 2020KY161).

### 2.2. Experimental Procedure

The experiment was conducted in a single-blind parallel design (Figure 2A). Participants visited the lab to receive a total of 30 min of active (20 Hz: N = 9 or 70 Hz: N = 10) or sham (no current was applied: N = 19) TI-tACS. An eye-closed resting state EEG was recorded for 2 min before and after each 10-min TI-tACS. The impedance and temperature of each stimulation electrode were monitored during the stimulation.

Before and after the stimulation, all the participants completed the neurological and neuropsychological measures listed below. Specifically, before the stimulation, NSE was measured. The participants then immediately completed several neuropsychological measurements, including MoCA [19], A-CalCAP [20], PPT [23] and VAMS-R [21,22] which were used to measure the mood state of participants. Participants also rated their states, including concentration, calmness, fatigue and visual perceptual changes, on a five-point self-assessment scale (SAS). After the stimulation, participants filled out SAS again then took another NSE test. Finally, the participants completed the rest of the measurements.

In addition, right after the last EEG recording, participants reported adverse effects (AEs) of the TI-tACS by filling out a subjective questionnaire [24,25], which included itching, headache, burning, warmth, tingling, metallic taste, fatigue, vertigo, nausea and phosphene. The extent of these sensations was rated from 0 to 4, representing none, mild, moderate, considerable and strong, respectively.

### 2.3. Neurological and Neuropsychology Tests

NSE, a sensitive biological marker of neuronal damage [18], was measured as a neurological outcome. In the current study, the NSE test was carried out in the nuclear medicine department of The First Affiliated Hospital of Anhui Medical University. The participants’ blood was drawn by nurses. In the test report, the NSE value and a reference value of the normal range (<16.3 ng/mL) are listed.

The MoCA [19] consists of a series of implementable cognitive assessments involving attention, executive function, memory, language, visual structure skills, conceptual thinking, computation and orientation. An alternative version was used in the post-test.

The CalCAP contains a series of computerized cognitive tasks to measure reaction time and the speed of information processing. The abbreviated version of CalCAP (A-CalCAP) [20] includes four tasks to assess psychomotor functions: (1) a simple reaction time task (SRT) to measure the basal reaction time, (2) choice reaction time for single digits (CRT) to add a simple process of memory, (3) serial pattern matching 1 (SPM1), where the participants were asked to respond only when the current displayed digit was the same as the previous one, and (4) serial pattern matching 2 (SPM2), where the participants were asked to respond only when the number was one more than the previous one.

The PPT [23] is a test of the dexterity of the unimanual and bimanual fingers and hands. It also contains four subtests. In the first three subtests, the participants were instructed to put as many pins as possible into the lengthwise arranged holes within 30 s with their right hand, left hand and both hands. In the last subtest, the participants were asked to construct as many “assemblies” as possible in 1 min with both hands alternately in a sequence manner. The order of the assembly is a pin, a washer, a collar and another washer. The tests were repeated three times to obtain reliable results.

### 2.4. EEG Recording

EEG was recorded with a UEA-16BZ amplifier (SYMTOP, Beijing, China). A 16-channel electrode cap (Greentek, Wuhan, China) with Ag/AgCl electrodes located according to the EEG international 10–20 system (Fp1, Fp2, F7, F3, F4, F8, Fz, C3, C4, Cz, P3, P4, Pz, O1, O2, Oz) was laid on the scalp. Reference electrodes were attached to the bilateral earlobes, and the ground electrode was located at AFz. C3 was not included in the recording electrode to avoid potential direct connection between stimulation electrodes mediated by conductive gel. The impedance between the reference and any recording electrode was kept under 10 kΩ. EEG signals were sampled at 1000 Hz, and then a low-pass filter with a cutoff frequency of 120 Hz and a 50 Hz notch filter were applied online. Participants were instructed to keep their eyes closed during the 2-min EEG recording. The EEG data were monitored by a clinical doctor (C.F.) and checked to assess any pathological activities, i.e., epileptic seizure, etc. Seizure activities were also scanned by automated epilepsy detection software (Encevis, AIT Austrian Institute of Technology GmbH, https://www.encevis.com/, accessed on 17 May 2022), which has been validated in several clinical datasets [27,28,29].

### 2.5. Temporal Interference Transcranial Alternating Current Stimulation

TI-tACS was applied by a customized battery-driven stimulator, whose performance was comparable with that of Grossman et al. [3,15]. For safety concerns, the stimulator is powered by batteries and isolated from the main electricity. The currents of all four output ports were monitored by protective circuits to ensure security. Once the amplitude of the current at any one output port exceeded the safety threshold, all four output ports were cut down by relays.

We used five circular Ag-AgCl electrodes with a radius of 1 cm (Pistim electrode, Neuroelectrics, Barcelona, Spain), four of which were stimulating electrodes, and one of which was the ground electrode located on the mastoid behind the participant’s left ear to avoid current accumulations for safety considerations. Stimulation electrodes were located around C3. Specifically, one pair of electrodes was located at FC3 and C5, and the other pair was located at CP3 and C1. The distance between the stimulation electrodes and C3 was 30 mm (Figure 2B). Electrodes were fixed by the electrode cap and filled with conductive gel (Quick-Gel, Compumedics USA Inc, Charlotte, NC, USA) to make the impedance of each electrode below 10 kΩ. The stimulation intensity was peak-to-peak 2 mA in a single channel (total peak-to-peak 4 mA for two channels). A simulation head model with the intensity of the electric field is shown in Figure 2C.

For the active stimulation, three blocks of 10-min TI-tACS were applied at 20 Hz (2000 Hz & 2020 Hz) or 70 Hz (2000 Hz & 2070 Hz). There were 30 s linear ramp up and ramp down periods at the beginning and the end of each stimulation block. Participants were asked to relax and keep their eyes open during the stimulation. For the sham stimulation, no current was applied to the participants during the three 10-min blocks, except for a 60 s ramp (30 s ramp up and 30 s ramp down) at the beginning of the block.

During the stimulation blocks, the stimulation waveforms were monitored by a hand-held oscilloscope (2C42, Hantek, Qingdao, China) during the stimulation, and the voltage of each electrode was recorded every 2 min to calculate the impedances. The temperature of each skin-electrode interface was monitored by a four-channel thermometer with T-type thermocouples (JK804, JINKO, Changzhou, China). The tips of the thermocouples were insulated to avoid direct skin contact. The TI-tACS stimulator and the oscilloscope were placed behind the participants, so they were blinded to the stimulation condition.

### 2.6. Statistical Analysis

All analyses were performed on MATLAB 2020a (MathWorks, Natick, MA, USA) and IBM SPSS Statistics 26.0 (IBM Corp, Armonk, NY, USA). Differences in changes in the neurological and neuropsychological measures before and after TI-tACS between the two groups (active vs sham) were assessed by 2 (group: active vs sham) × 2 (testing time: before TI-tACS vs after TI-tACS) repeated measures analysis of variance (rmANOVA). Group differences in AEs were assessed by the chi-square test.

Thirty-three participants’ EEG data (active group: 17, sham group: 16) were involved in the following statistical analyses, since 5 participants’ EEG data were contaminated and were excluded. The power spectral density (PSD) calculation was performed in MATLAB using EEGLAB Toolbox [30] after bandpass filtering between 0.1 and 48 Hz. The spectra were then divided into six frequency bands, namely, delta (1–4 Hz), theta (4–8 Hz), alpha (8–13 Hz), low beta (13–20 Hz), high beta (20–30 Hz) and low gamma (30–45 Hz). The means of each band were calculated to represent the band power. Finally, 2 (group: active vs sham) × 4 (recording time: before TI-tACS, after the first stimulation, after the second stimulation, after the third stimulation) rmANOVA was performed to test the group differences in changes in each channel and each frequency band.

Sample size was determined by G∗Power 3.1 [31] for rmANOVA based on previous transcranial electrical stimulation studies [32,33,34], with an effect size of 0.3 (Cohen’s d), a power of 0.9 and an α error probability of 0.05. The calculated total sample size was 32. Considering potential dropouts, we recruited slightly more participants.

## 3. Results

No significant differences in the neurological (NSE) or neuropsychological (MoCA, A-CalCAP, PPT, VAMS-R, SAS) measurements were detected (more details are given in Table 1). The relative change percentages (100 * pre-test / post-test) of these measurements were also checked, and no significant difference was found (more details are given in Appendix A). All the NSE results are in the normal range. In addition, there was no significant difference in changes in EEG band powers between the active group and the sham group (more details are given in Appendix A). Automated epilepsy detection by software also detected no epileptic activity in EEG data from the two groups. Temperatures measured at the skin-electrode interfaces were all below body temperature (range 25.6–35.3 °C, median at 29.8 °C). The lowest impedance in one active stimulation block was 353.33 ± 22.25 Ω (mean ± SD), larger than the internal body resistance (300 Ω).

AEs during TI-tACS were minor and tolerable according to the participants’ descriptions and our observations. Specifically, among the 38 participants, only three reported more than moderate discomfort (one in the active group and two in the sham group, *χ*^2^ = 0.362, *p* = 0.547). Although a few participants reported mild adverse effects, there was no significant difference in any subitem of AEs between the active group and sham group (more details are given in Table 2). Meanwhile, no serious adverse events, such as injury, were reported or observed in the active stimulation group of 19 participants. In addition, it is a well-conducted single-blind experiment, since only 47.4% of participants identified their stimulation condition correctly (chance level: 50%).

## 4. Discussion

In this study, we evaluated the safety of TI-tACS by applying TI-tACS to healthy human participants in conditions similar to typical TES conditions (2 mA, 20/70 Hz, 30 min). We investigated the neurological (NSE) and neuropsychological (MoCA, A-CalCAP, PPT, VAMS-R, SAS) measures before and after TI-tACS and found no significant change compared to the sham stimulation group. The EEG signals were also recorded before and after each stimulation block, and no significant change in band power or pathological activity, such as epileptic seizures, was found. Furthermore, most participants reported no discomfort during the experiment.

Since 2017, Grossman et al. brought TI-tACS into the field of brain stimulation [3], and many subsequent studies have shown evidence supporting its effectiveness [6,7,8,9,10,11,12,13,14,15,16]. However, to the best of our knowledge, this is the first study to test the safety of TI-tACS in the human brain. In regard to safety, Grossman et al. checked the safety characterization of TI-tACS in mouse experiments and found that TI-tACS did not cause neuronal apoptosis, DNA damage, neuroinflammatory reactions or changes in synapse density. They also found that high-frequency electric fields did not induce significant temperature changes in mouse brains [3]. Similarly, in our human experiments, we found no significant difference in the changes in any of the neurological and neuropsychology measures before and after TI-tACS between the active and sham groups. These results indicate that TI-tACS did not cause any neuronal damage, cognitive impairment, impairment of dexterity in the hands or significant changes in mood or other psychological states. In addition, the temperature of all the skin-electrode interfaces was below body temperature, which indicates that TI-tACS did not heat or burn the skin. The impedance of all the stimulation electrodes was above the internal body resistance, indicating that the skin barrier function was not breached during TI-tACS. With these safety results of mouse and human experiments, we argue that TI-tACS is safe enough to be employed in human studies.

We also found some limitations in our study that need to be overcome in future studies. The current study only verified the safety of TI-tACS in the specified conditions without exploring the safety in other conditions. Therefore, future TI-tACS studies with higher current intensities, higher current frequencies or longer stimulation durations still need to test their safety. In the current study, the evidence supporting the results of the risk of adverse events of TI-tACS is weak, since there were only 19 participants in the active stimulation group. To achieve more robust results, more participants in active stimulation need to be recruited in future studies. Although no serious adverse events were found in the current study, serious adverse events, such as potential life-threatening conditions, should be investigated with a larger sample and a longer follow-up in future studies. Meanwhile, as a global brain injury measurement, NSE may not be sensitive enough to detect local injury from focused brain stimulation. Therefore, future studies may employ structural magnetic resonance imaging (MRI) to measure potentially more circumscribed brain injuries as a complement to NSE. Furthermore, to achieve a more credible result, follow-up surveys are recommended in future studies.

## 5. Conclusions

This study experimentally supported that TI-tACS is safe and tolerable for humans in conditions similar to typical TES conditions (2 mA, 20/70 Hz, 30 min). No evidence indicated that TI-tACS may induce any neurological or neuropsychological state changes and no serious or intolerable adverse effects were found in the current study. This study lays the foundation for future human studies and clinical studies with TI-tACS.

## Figures and Tables

**Figure 1 brainsci-12-01194-f001:**
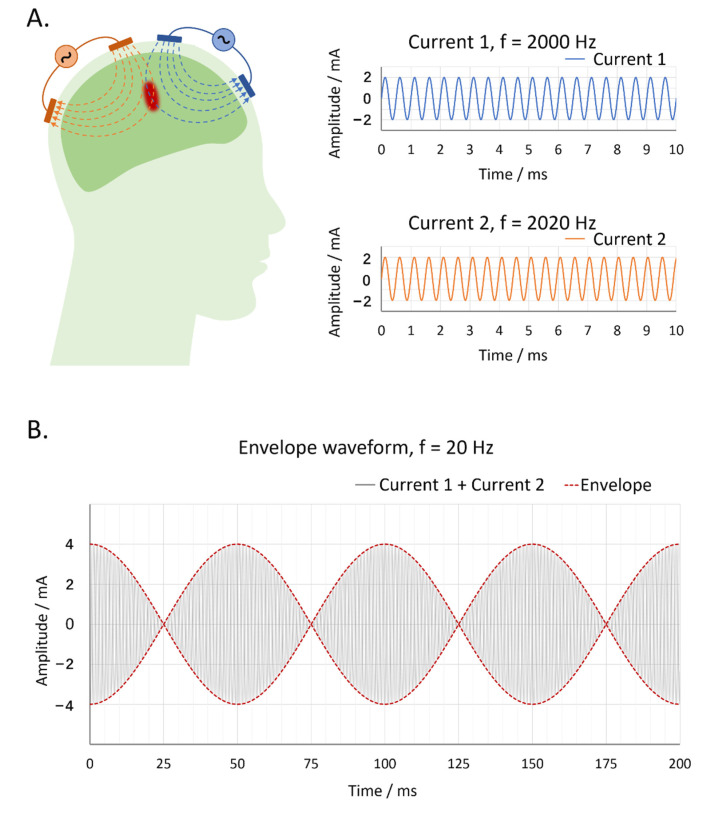
Schematic diagram of TI-tACS. (**A**) Stimulation of a specific area in the deep brain by TI-tACS, which has two pairs of electrodes with high-frequency alternative currents (e.g., 2000 Hz and 2020 Hz). (**B**) Low-frequency envelope-modulated current waveform of TI-tACS (e.g., 20 Hz) that is generated by the superposition of the two high-frequency current waves shown in (**A**). TI-tACS: temporal interference transcranial alternating current stimulation.

**Figure 2 brainsci-12-01194-f002:**
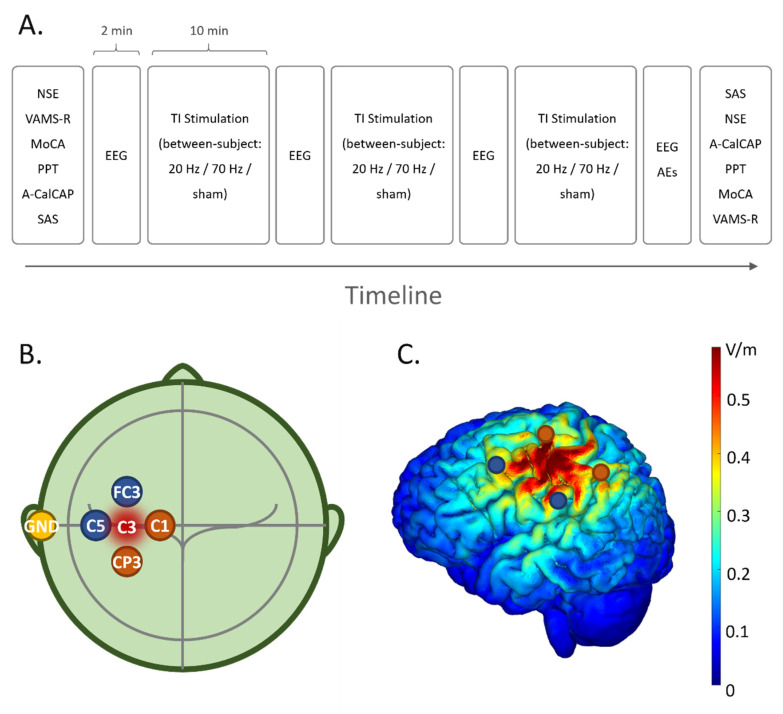
(**A**) Experimental design. Before and after the stimulation, participants completed the neurological (NSE) and neuropsychological (MoCA, A-CalCAP, PPT, VAMS-R, SAS) measures. Participants received 3 blocks of 10-min active (20 Hz: N = 9 or 70 Hz: N = 10) or sham (no current was applied: N = 19) TI-tACS. An eye-closed resting state electroencephalograph (EEG) was recorded for 2 min immediately before and after a 10-min TI-tACS. During the TI-tACS, participants wore the EEG cap all the time. After the last EEG recording, participants reported adverse effects (AEs) of TI-tACS by filling out a subjective questionnaire. The order of VAMS-R, MoCA, PPT and A-CalCAP was balanced among participants and the order of the post-test was inverse to that of the pre-test in each participant. (**B**) Location of stimulation electrodes and the target region. Based on the international 10–20 system, to stimulate the left primary motor cortex (C3), two pairs of stimulation electrodes (FC3, C5; CP3, C1) were placed approximately 30 mm around the target region. Additionally, the ground electrode (GND) was placed at the left mastoid. (**C**) Simulation head model with the intensity of the electric field. NSE: Serum neuron-specific enolase; MoCA: Montreal Cognitive Assessment; A-CalCAP: an abbreviated version of the California Computerized Assessment Package; PPT: Purdue Pegboard Test; VAMS-R: revised version of the Visual Analog Mood Scale; SAS: self-assessment scale.

**Table 1 brainsci-12-01194-t001:** The statistical results of neurological and neuropsychological measurements.

Measurements (Range or Unit)	Active Group (Mean ± SD)		Sham Group (Mean ± SD)		Statistical Results	
	Pre	Post	Pre	Post	*F*	*p*
MoCA (0–30)	27.95 ± 1.47	27.68 ± 1.46	27.63 ± 1.17	27.89 ± 1.49	0.973	0.331
PPT (times ^1^)						
Right Hand	16.44 ± 1.56	17.21 ± 1.91	16.35 ± 1.60	17.65 ± 0.97	1.985	0.167
Left Hand	15.25 ± 1.86	16.10 ± 2.06	14.93 ± 1.29	15.91 ± 1.42	0.298	0.588
Both Hands	12.72 ± 1.68	13.35 ± 1.68	12.51 ± 1.60	13.30 ± 1.62	0.307	0.583
Assembly	41.30 ± 5.88	45.84 ± 6.68	40.23 ± 7.75	44.14 ± 6.40	0.340	0.564
A-CalCAP (ms ^2^)						
SRT	363.79 ± 82.52	357.17 ± 60.90	357.64 ± 50.01	367.96 ± 54.43	0.620	0.436
CRT	428.42 ± 33.54	441.08 ± 41.93	415.76 ± 35.18	422.66 ± 32.58	0.349	0.558
SPM1	501.01 ± 69.28	513.59 ± 83.14	487.74 ± 61.24	481.20 ± 65.78	1.589	0.216
SPM2	593.20 ± 94.59	540.83 ± 78.03	554.99 ± 67.63	535.68 ± 68.15	2.886	0.098
NSE (ng/mL)	14.09 ± 3.21	16.01 ± 2.94	13.40 ± 3.27	14.32 ± 3.72	0.460	0.503
VAMS-R (0–100)						
Sad	1.74 ± 2.16	8.63 ± 24.43	6.26 ± 9.15	9.21 ± 15.51	0.418	0.522
Confused	9.00 ± 19.06	8.16 ± 22.60	15.74 ± 18.47	11.00 ± 17.02	0.462	0.638
Afraid	7.05 ± 22.69	6.32 ± 21.76	4.53 ± 9.06	5.68 ± 11.33	0.060	0.808
Happy	47.00 ± 34.39	43.89 ± 34.41	49.26 ± 31.58	49.37 ± 27.95	0.098	0.756
Tired	31.21 ± 32.54	35.16 ± 30.03	35.47 ± 34.30	31.47 ± 29.62	0.928	0.342
Angry	7.05 ± 22.41	9.68 ± 25.74	2.53 ± 4.61	5.42 ± 10.60	0.006	0.941
Tense	16.68 ± 30.35	8.05 ± 22.31	5.74 ± 8.85	8.89 ± 20.32	2.361	0.133
Energetic	49.21 ± 28.49	47.84 ± 28.75	60.00 ± 30.01	49.16 ± 30.87	1.692	0.202
SAS (1–5)						
Concentration	3.74 ± 0.73	3.26 ± 0.81	3.32 ± 0.48	3.21 ± 0.63	3.196	0.082
Calmness	4.11 ± 0.74	3.79 ± 0.92	3.68 ± 0.75	3.58 ± 0.84	0.475	0.495
Fatigue	2.58 ± 0.84	3.05 ± 1.03	2.37 ± 0.96	3.16 ± 0.96	1.317	0.259
Visual perception	3.68 ± 0.67	3.37 ± 0.60	3.37 ± 0.90	3.32 ± 1.00	1.573	0.218

The values shown in the table are the descriptive measures and statistical results of 2 groups × 2 recording times repeated measures analysis of variance (rmANOVA) of each neurological and neuropsychological measurement. All statistical significances are larger than 0.05. ^1^ The values are the numbers of times that the participants completed in set time. ^2^ The values are the mean reaction times.

**Table 2 brainsci-12-01194-t002:** Reported adverse effects (AEs) after TI-tACS and their statistical results.

Items	Active Group	Sham Group	*χ^2^*	*p*
Itching	mild: 2	none	2.111	0.146
Headache	none	mild: 3	3.257	0.071
Burning	none	none	- ^1^	-
Warmth	mild: 1	mild: 1	0.000	1.000
Tingling	mild: 1	none	1.027	0.311
Metallic taste	none	none	-	-
Fatigue	mild: 2; considerable: 1	mild: 2; moderate: 2	0.175	0.676 ^2^
Vertigo	moderate: 1 ^3^	mild: 4	2.073	0.150
Nausea	none	none	-	-
Phosphene	none	none	-	-

The values shown in the table are the descriptive measures and statistical results of the chi-square test between the two groups. All statistical significances are larger than 0.05. ^1^ Not meeting the requirements of chi-square test. ^2^ The statistical result of fatigue is calculated with the total headcounts of each group, without differentiating the extent of the effect. ^3^ This is the same participant who felt considerable fatigue.

## Data Availability

Data are available upon reasonable request to the corresponding author.

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
