# Peer review of "Safety Evaluation of Employing Temporal Interference Transcranial Alternating Current Stimulation in Human Studies"

_brainsci, 2022, doi:10.3390/brainsci12091194_

Round 1

Reviewer 1 Report

The manuscript is an interesting study exploring the effect of TI-ACS on EEG, neural biomarkers, and behaviors in healthy adults.The technique is potentially important, however, no effect was found between real and sham stimulation. For publishing the present paper, the authors may have to explain the potential utility of this new technology in the discussion.

Some comments are attached as below:

Abstract:

Its effectiveness has been verified in mouse, simulation and human studies. Please consider rephasing

“neurological and neuropsychological measurements” what are the measurements?

Methods:

What were the subjects doing when they received tACS?

Flowchart: time for setting up EEG after TACS? Or the subjects wear an EEG cap during tACS? The order of those assessments needs to be specified.

NSE: please describe the procedure for taking this biomarker.

MoCA: alternative version was used during repeated measures?

Results:

Table 2: What were the p values?

Table 1& 3, please report the statistical value (eg. F value) along with the p values.

Conclusion

I do not think it is suitable to say tACS is a form of deep brain stimulation.

Reviewer 2 Report

This is an interesting study,

However, there are some important major points to be addressed.

-          Why this technique is called Temporal Interference tACS? What is the temporal factor and how can be demonstrated that this is an interference. If the frequency is in the range of 1KHz that does not necessary mean that may create an interference in cortical and sub-cortical (in the case indirectly reaches deeper structures) regions.

-          The mechanisms of how the difference of the two electrical fields can induces a low frequency stimulation is very unclear. The best way would be to describe the difference between TI-tACS and conventional tACS. Without a more detailed technical description of the technique the current manuscript cannot be suitable for publication.

-          It would be very important whether is possible to normalize post-effect versus pre-effect in each group. For example, by dividing post data versus pre data and multiply by 100 as a percentage. Once the authors have just single values for each test, then they could compare SHAM versus TI-tACS, to check if there is a real difference between the 2 conditions. This is because the mere differences between pre and post cannot guarantee a reliable change due by the stimulation. There may be too many factors that may affect post-sham or post-TI-tACS.

-          Again, it is fundamental to describe this novel technique in more details. How this differs from conventional TES techniques such as tACS.

-          Would be important to mention that Transcranial Magnetic Stimulation would be ideal technique to measure offline and online TI-tACS effects since the tACS studies on motor cortex helped to reveal the important mechanism of the sinusoidal stimulation.

Reviewer 3 Report

This manuscript is aiming to explore the safety of temporal interference tACS in humans. The authors included EEG measures, NSE (as a marker of brain injury), as well as global cognitive measures, and questionnaires on self-reported changes in mood and other potential side effects. They compared two difference active protocols (20Hz vs 70Hz) to sham and did not find differences in any of the above measures. This manuscript is the first paper to evaluate safety in human subjects using multiple measures. The presentation is clear and straightforward. The experimental design is appropriate. They concluded that temporal interference tACS protocol is safe for application in human subjects. I only have a few minor suggestions and comments.

1. Would need more information on how NSE is measured and please provide normal range per lab standard (Line 113). Also any participants took medications? Since some medicine such as proton pump inhibitor may affect its level.

2. Please elaborate how the polarity of current is alternated among the stimulating electrodes to create the electric field (Line 184-185)

3. Would be helpful if there is simulation head model of how deep this electric field may reach and the intensity of the electric field to help understand how much dose is given (Figure 2). Also there is misspelling of FC3 (spelled as CF3) in Figure 2.

4. Would help readers to interpret the measures if the score ranges (and units) could be provided for MoCA, A-CalCAP, PPT, VAMS-R, and SAS (for Table 2).

5. NSE is generally used for more global brain injury and may not be sensitive enough to detect local injury from focused brain stimulation. Other measures such as structural MRI (such as diffusion imaging) may be complementary to NSE in future research for measuring potentially more circumscribed brain injury.

6. May also need to be more conservative in mentioning that TI-tACS protocols with more sessions (than 30 minutes) and higher intensity (than 2 mA) will still need to be examined for safety issues, since dosage may accumulate overtime to cause injury.

Round 2

Reviewer 1 Report

My comments have been well addressed.

Author Response

Dear professor,

Thanks a lot for your helpful suggestions. Your suggestion make our manuscript much better.

best regards,

The authors